# RAGE–TLR4 Crosstalk Is the Key Mechanism by Which High Glucose Enhances the Lipopolysaccharide-Induced Inflammatory Response in Primary Bovine Alveolar Macrophages

**DOI:** 10.3390/ijms24087007

**Published:** 2023-04-10

**Authors:** Longfei Yan, Yanran Li, Tianyu Tan, Jiancheng Qi, Jing Fang, Hongrui Guo, Zhihua Ren, Liping Gou, Yi Geng, Hengmin Cui, Liuhong Shen, Shumin Yu, Zhisheng Wang, Zhicai Zuo

**Affiliations:** 1College of Veterinary Medicine, Sichuan Agricultural University, Chengdu 611134, China; 2Institute of Animal Nutrition, Sichuan Agricultural University, Chengdu 611134, China

**Keywords:** TLR4–RAGE crosstalk, glucose, lipopolysaccharide (LPS), inflammatory, alveolar macrophages

## Abstract

The receptor of advanced glycation end products (RAGE) and Toll-like receptor 4 (TLR4) are important receptors for inflammatory responses induced by high glucose (HG) and lipopolysaccharide (LPS) and show crosstalk phenomena in inflammatory responses. However, it is unknown whether RAGE and TLR4 can influence each other’s expression through a crosstalk mechanism and whether the RAGE–TLR4 crosstalk related to the molecular mechanism of HG enhances the LPS-induced inflammatory response. In this study, the implications of LPS with multiple concentrations (0, 1, 5, and 10 μg/mL) at various treatment times (0, 3, 6, 12, and 24 h) in primary bovine alveolar macrophages (BAMs) were explored. The results showed that a 5 μg/mL LPS treatment at 12 h had the most significant increment on the pro-inflammatory cytokine interleukin 1β (IL-1β), IL-6, and tumor necrosis factor (TNF)-α levels in BAMs (*p* < 0.05) and that the levels of TLR4, RAGE, MyD88, and NF-κB p65 mRNA and protein expression were upregulated (*p* < 0.05). Then, the effect of LPS (5 μg/mL) and HG (25.5 mM) co-treatment in BAMs was explored. The results further showed that HG significantly enhanced the release of IL-1β, IL-6, and TNF-α caused by LPS in the supernatant (*p* < 0.01) and significantly increased the levels of RAGE, TLR4, MyD88, and NF-κB p65 mRNA and protein expression (*p* < 0.01). Pretreatment with FPS-ZM1 and TAK-242, the inhibitors of RAGE and TLR4, significantly alleviated the HG + LPS-induced increment of RAGE, TLR4, MyD88, and NF-κB p65 mRNA and protein expression in the presence of HG and LPS (*p* < 0.01). This study showed that RAGE and TLR4 affect each other’s expression through crosstalk during the combined usage of HG and LPS and synergistically activate the MyD88/NF-κB signaling pathway to promote the release of pro-inflammatory cytokines in BAMs.

## 1. Introduction

Long-distance transportation is an essential part of the beef cattle industry, but transportation stress caused by long-distance transportation can weaken cattle’s immune systems causing the occurrence of bovine respiratory disease complex (BRDC) [1]. During the occurrence of BRDC, gram-negative bacterial infections are a major cause of pneumonia [2,3]. Lipopolysaccharide (LPS) and hyperglycemia were considered the most common contributors to the pulmonary inflammatory response [4]. LPS, the primary element of gram-negative bacteria’s outer membrane, can activate alveolar macrophages (AMs) and promote the release of pro-inflammatory cytokines such as IL-1β, IL-6, and TNF-α [5,6]. Furthermore, transportation stress elevates blood glucose in cattle, leading to hyperglycemia [7,8]. Hyperglycemia is not only related to the release of pro-inflammatory cytokines in AMs, such as IL-1β, IL-6, and TNF-α [9,10,11], but is also associated with the production of a cytokine storm [12]. Several studies have reported that HG and LPS promote AM activation and that HG can exacerbate LPS-induced inflammatory response [4,13,14], but the molecular mechanism needs to be further investigated.

TLR4 is an intrinsic immune receptor that is closely associated with inflammatory responses and cytokine storm formation [15,16]. It is the receptor for LPS in triggering myeloid differentiation factor 88 (MyD88) and nuclear factor kappa-B (NF-κB) inflammatory signaling pathways and promoting the production and release of pro-inflammatory cytokines IL-1β, IL-6, and TNF-α [17,18,19,20]. Dasu et al. [21] found that HG-induced upregulation of Toll-like receptor 2 (TLR2) and TLR4 expression in human monocytes further promoted the activation of NF-κB. Mudaliar et al. [22] reported that inhibition of TLR2 or TLR4 significantly inhibited HG-induced NF-κB activation and the synthesis and release of pro-inflammatory cytokines and chemokines in human microvascular endothelial cells. These studies suggested that TLR4 is also closely associated with the HG-induced inflammatory response. 

RAGE, another important receptor involved in the inflammatory response, can bind with various ligands, including advanced glycosylation end products (AGEs) [23], high mobility group box 1 (HMGB1) [24], and the S100A8/A9 heterodimer [25]. The ligands mentioned above are increased in the HG-induced inflammatory response [26,27]. HG induces significant upregulation of RAGE and its ligands, which then bind to RAGE to enhance RAGE oligomerization and further activate the cellular inflammatory signaling pathways [27]. Wang et al. [28] found that the use of RAGE-blocking antibodies effectively inhibited LPS-induced NF-κB activation in endothelial cells, suggesting that RAGE plays an important role in the LPS-induced inflammatory response. 

Previous studies found that RAGE and TLRs have a similar mechanism of action and similar ligands and inflammatory signaling pathways are shared by TLR4 and RAGE, which have a synergistic activation effect on downstream signaling pathways when both receptors are activated. The above phenomenon is called “cross-talk” or crosstalk [29,30]. Specifically, upon activation of RAGE and TLR4, downstream signaling pathways of the two receptors converge and activate synergistically at multiple levels. The use of TLR4-related signaling pathways to transduce activation signals from RAGE can activate a wider range of signaling pathways as well as cause more intense signaling after RAGE binds to ligands [29,31]. Both HG and LPS can activate RAGE and TLR4 and share downstream inflammatory signaling pathways. However, it is unclear whether RAGE–TLR4 crosstalk is involved in the combination treatment of HG and LPS-induced inflammation response in BAMs. In addition, whether RAGE interacts with TLR4 to influence each other’s expression at the receptor level and synergistic activation of the MyD88/NF-κB pathway in the inflammatory response of BAMs needs to be further investigated. 

In this experiment, we aimed to investigate the role of RAGE–TLR4 crosstalk in the activation of the MyD88/NF-κB signaling pathway and release of pro-inflammatory cytokines TNF-α, IL-1β, and IL-6 in BAMs induced by the combination of HG and LPS. We used FPS-ZM1 (a RAGE inhibitor) and TAK-242 (a TLR4 inhibitor) to confirm the involvement of the RAGE–TLR4 crosstalk in this inflammatory response. Our results suggest that the crosstalk between RAGE and TLR4 plays a critical role in the inflammatory response induced by the combination of HG and LPS in BAMs. These findings highlight the potential therapeutic significance of targeting the RAGE–TLR4 crosstalk in the treatment of inflammation-related diseases. 

## 2. Results

### 2.1. LPS Increased Pro-Inflammatory Cytokine Release in Primary BAMs in a Dose- and Time-Dependent Manner

BAMs were treated with LPS at a range of various doses (0, 1, 5, and 10 μg/mL) for 12 h. The results are depicted in Figure 1A–C. Contrasted with the control group (0 μg/mL), all three LPS concentrations significantly raised the levels of IL-1β, IL-6, and TNF-α (*p* < 0.05). In addition, 5 μg/mL of LPS showed the strongest increment in the release of IL-6 and TNF-α (*p* < 0.01) compared to 1 μg/mL and 10 μg/mL LPS treatment groups. In comparison to the control group, the 10 μg/mL LPS treatment significantly raised the levels of IL-1β, IL-6, and TNF-α (*p* < 0.05), but there was no significant difference with the 1 μg/mL LPS treatment (*p* > 0.05). Therefore, we selected 5 μg/mL as the LPS concentration for subsequent experiments.

Similarly, the BAMs were exposed to 5 μg/mL LPS at a series of times (0, 3, 6, 12, and 24 h), and the results are displayed in Figure 1D–F. The levels of IL-1β and IL-6 significantly raised from the 6th h (*p* < 0.05) and reached their highest values at the 12th h and then reduced (*p* < 0.05) (Figure 1D,E). In addition, the TNF-α level significantly increased from the 3rd h (*p* < 0.05), reached its peak value at the 12th h, and then maintained. Therefore, we selected 12 h as the treatment duration for the following trials.

### 2.2. TLR4, RAGE, and Their Interaction Were Involved in the Inflammatory Response Caused by LPS

The levels of TLR4 and RAGE mRNA expression in the BAMs were tested, and the results were shown in Figure 2A,B,D,E. The levels of TLR4 and RAGE mRNA expression were dramatically elevated in groups with LPS treatment (*p* < 0.05), and the 5 μg/mL LPS treatment group showed the most significant effects (*p* < 0.01). Intriguingly, there was no effect on the levels of TLR4 mRNA expression from 10 μg/mL LPS treatment (Figure 2A). In addition, the levels of RAGE and TLR4 mRNA expression significantly raised from the 6th h (*p* < 0.05) and reached the peak value at the 24th h after LPS treatment (*p* < 0.01) (Figure 2B,E). 

The BAMs were pretreated with 10 μM TLR4 inhibitor TAK-242 and 1.0 μM RAGE inhibitor FPS-ZM1 for 1 h, respectively, and dimethyl sulfoxide (DMSO) treatment performed as the solvent control group. According to Figure 2C,F, in the FPS-ZM1 pretreatment group, not only did the level of RAGE mRNA expression significantly reduce but the level of TLR4 mRNA expression also significantly decreased (*p* < 0.01). Similarly, in the TAK-242 pretreatment group, the levels of TLR4 and RAGE mRNA expression also significantly decreased (*p* < 0.01). The results of the levels of RAGE and TLR4 protein expression are displayed in Figure 2G. The levels of TLR4 and RAGE protein expression were considerably raised in the LPS treatment group (*p* < 0.01). As a result of the FPS-ZM1 pretreatment, the levels of RAGE and TLR4 protein expression in the LPS treatment group were significantly reduced (*p* < 0.01), and the TAK-242 pretreatment showed the same results (*p* < 0.01). The mRNA and protein levels of RAGE decreased when TLR4 expression was inhibited with TAK-242, and the mRNA and protein levels of TLR4 decreased when RAGE expression was inhibited with FPS-ZM1, suggesting crosstalk between RAGE and TLR4. Then, the levels of IL-1β, IL-6, and TNF-α were detected, and the results are displayed in Figure 3H–J. Pretreatment with 1.0 μM FPS-ZM1 and 10 μM TAK-242 both significantly decreased the levels of IL-1β, IL-6, and TNF-α in BAMs (*p* < 0.01). 

### 2.3. RAGE and TLR4 Synergistically Activate the MyD88/NF-κB Signaling Pathway in the Inflammation Response Caused by LPS

The levels of MyD88 and NF-κB p65 mRNA and protein expression were detected. Figure 3A,B,D,E showed that the 5 μg/mL and 24 h of LPS treatment group showed the most significantly increased levels of MyD88 and NF-κB p65 mRNA expression compared with the control group (*p* < 0.01). Similarly, in the inhibitor experiments, pretreatment with FPS-ZM1 and TAK-242 blocked the LPS-induced increase in the levels of MyD88 and NF-κB P65 mRNA expression (*p* < 0.01) (Figure 4C,F). The levels of MyD88 and NF-κB P65 protein expression were similar in the DMSO and control groups (*p* > 0.05), but, compared to the control group, they are significantly increased in the LPS treatment group (*p* < 0.01) (Figure 3G). However, in contrast to the LPS treatment group, pretreatment with FPS-ZM1 and TAK-242 significantly blocked the LPS-induced increases in MyD88 and NF-κB P65 protein expression levels (*p* < 0.01) (Figure 3G,H).

### 2.4. HG Enhanced LPS-Induced Pro-Inflammatory Cytokine Secretion and Upregulated the RAGE/TLR4/MyD88/NF-κB p65 Pathway in BAMs

The levels of RAGE, TLR4, MyD88, and NF-κB p65 mRNA expression in the HG group and normal glucose (NG, 5.5 mM) group were assessed, and Figure 4A–D displays the result. At 1 μg/mL and 5 μg/mL LPS stimulus, compared to the NG group, the levels of RAGE, TLR4, MyD88, and NF-κB p65 mRNA expression in the HG group were significantly increased (*p* < 0.01). Under 10 μg/mL LPS treatment, the levels of *NF-κB p65* (*p* < 0.05) and *RAGE* (*p* < 0.01) mRNA expression in the HG group were significantly increased compared with the NG group, but the levels of TLR4 and MyD88 mRNA expression were comparable to those of the NG group (*p* > 0.05). Additionally, the levels of IL-1β, IL-6, and TNF-α in the HG and NG groups were tested, and Figure 4E–G showed the results. After 1 and 5 μg/mL LPS treatment, the levels of IL-1β, IL-6, and TNF-α in the HG group were significantly higher than those in the NG group (*p* < 0.05). Under 10 μg/mL LPS treatment, the level of TNF-α in the HG group was significantly raised (*p* < 0.01) compared with the NG group, although the levels of IL-1β and IL-6 were comparable to those of the NG group (*p* > 0.05).

### 2.5. RAGE–TLR4 Crosstalk Regulated the Synergism between HG and LPS on the Inflammatory Response in BAMs

The levels of RAGE, TLR4, MyD88, and NF-κB p65 mRNA expression in each group were detected, and the results are displayed in Figure 5A–D. In the HG group or LPS group, the levels of RAGE, TLR4, MyD88, and NF-κB p65 mRNA expression were significantly raised compared with the control group (*p* < 0.05) and were significantly lower than in the HG + LPS group (*p* < 0.01). Similarly, pretreatment with FPS-ZM1 and TAK-242 significantly blocked the HG + LPS-induced increase in RAGE, TLR4, MyD88, and NF-κB p65 mRNA expression in the HG + LPS group (*p* < 0.01). The levels of RAGE, TLR4, MyD88, and NF-κB p65 protein expression in each group were also detected, and Figure 5E–J displays the results. The levels of RAGE, TLR4, MyD88, and NF-κB p65 protein expression in the HG or LPS group were significantly increased compared to the control group (*p* < 0.05) and were significantly reduced than those in the HG + LPS group (*p* < 0.01). In the HG + LPS group, pretreatment with FPS-ZM1 or TAK-242 significantly inhibited the HG + LPS-induced increase in RAGE, TLR4, MyD88, and NF-κB p65 protein expression (*p* < 0.01). Figure 5K–M displayed the results of IL-1β, IL-6, and TNF-α levels in each group. The levels of IL-1β, IL-6, and TNF-α in the DMSO group were comparable to the control group (*p* > 0.05). The levels of IL-1β, IL-6, and TNF-α levels in the HG group and LPS group were significantly increased (*p* < 0.01) compared to the control group but considerably reduced in contrast to the HG + LPS group (*p* < 0.01). Pretreatment with FPS-ZM1 and TAK-242 significantly ameliorated the HG + LPS-induced increase in IL-1β, IL-6, and TNF-α levels in the HG + LPS group (*p* < 0.01).

## 3. Discussions

Stress-induced hyperglycemia and gram-negative bacterial infections are two of the primary factors contributing to severe pulmonary inflammation of bovine during long-distance transportation [1,32]. RAGE and TLR4 are the main receptors for LPS and HG leading to the development of cellular inflammatory responses, and RAGE–TLR4 crosstalk occurs when RAGE and TLR4 are activated, thereby synergistically activating shared downstream inflammatory signaling pathways [31]. However, it is not clear whether RAGE–TLR4 crosstalk occurs during HG and LPS co-treatment-induced inflammation in BAMs, and the effects of RAGE–TLR4 crosstalk on downstream-related signaling pathways and specific mechanisms need to be further elucidated. Previous studies have demonstrated that LPS can promote the synthesis and release of pro-inflammatory cytokines, such as TNF-α, IL-1β, and IL-6, from a variety of cell types, including placental cells, endothelial cells, and macrophages [33,34,35]. In our study, the levels of pro-inflammatory cytokines IL-1β, IL-6, and TNF-α in the supernatant considerably elevated with the increased concentration of LPS and the duration of treatment, indicating that LPS induces BAMs to release pro-inflammatory cytokines in a dose- and time-dependent manner.

LPS promotes the release of pro-inflammatory cytokines from cells associated with various transmembrane receptors on the cell surface, mainly TLR4 [18] and RAGE [28]. We assumed that the LPS-induced inflammatory response in BAMs via receptor crosstalk occurred when RAGE and TLR4 were simultaneously activated. In our work, the levels of TLR4 and RAGE expression were increased after LPS treatment. However, TAK-242 (TLR4 inhibitor) and FPS-ZM1 (RAGE inhibitor) inhibited TLR4 and RAGE expression as well as the release of IL-1β, IL-6, and TNF-α in BAMs. These results indicated that both RAGE and TLR4 are involved in LPS-induced inflammatory response in BAMs. Intriguingly, we found that the inhibition of RAGE could down-regulate the level of TLR4 expression, and inhibition of TLR4 could also down-regulate the level of RAGE expression, indicating that RAGE and TLR4 mutually affect each other’s expression at the receptor level in the inflammatory response of BAMs caused by LPS. Prior research has demonstrated that LPS could trigger RAGE and TLR4 and cause RAGE–TLR4 crosstalk [36]. 

MyD88, a classical downstream signaling adaptor molecule of TLR4, is important for triggering TLR4-related inflammatory signaling pathways [37]. However, some studies have found that MyD88 could bind to RAGE and transduce a signal to downstream molecules, blocking the function of MyD88 abrogated intracellular signaling from HMGB1-activated RAGE [38], but the study had shortcomings. The HMGB1 used in the study had been shown to bind not only to RAGE but also to TLR4; so, blocking MyD88 may cause signal attenuation through the TLR4 pathway rather than the RAGE pathway. It is still unclear whether MyD88 can be coordinately regulated by RAGE and TLR4. The binding of LPS to TLR4 transmits inflammatory signals intracellularly through the MyD88-independent (TRAM) and MyD88-dependent pathways, activating NF-κB and promoting the synthesis and release of multiple pro-inflammatory cytokines [39,40]. Gąsiorowski et al. [31] found that RAGE and TLR4 exert synergistic activation on common downstream signaling pathways, such as NF-κB and AP-1. In the present study, the levels of MyD88 and NF-κB p65 expression were significantly upregulated after LPS treatment. However, TAK-242 and FPS-ZM1 significantly decreased the levels of MyD88 and NF-κB p65 expression. These results indicated that RAGE and TLR4 synergistically activate the MyD88/NF-κB signaling pathway in LPS-induced inflammatory responses of BAMs. With similar results to ours, Byrd et al. [41] found that LPS induced upregulation of TLR4, MyD88, NF-κB expression and pro-inflammatory cytokine TNF-α release in primary mouse macrophages. Wang et al. [28] also found that LPS caused the upregulation of RAGE protein expression and activated NF-κB, while inhibition of RAGE inhibited LPS-induced NF-κB activation in venous endothelial cells.

Glucose is an important source of energy for mammals, and NG (mostly 5.5 mM in in vitro experiments) ensures the energy demand and utilization of cells [29]. However, when glucose concentration is elevated (mostly 25.5 mM in in vitro experiments), it has a pro-inflammatory response to cells [42,43]. In our study, the co-treatment of LPS and HG caused a significant increase in the levels of RAGE, TLR4, MyD88, and NF-κB p65 mRNA expression and the release of IL-1β, IL-6, and TNF-α. The results indicated that HG enhanced the LPS-induced inflammatory response through further activation of the RAGE/TLR4/MyD88/NF-κB p65 signaling pathway in BAMs. Consistent with our results, Nielsen et al. [44] and Kong et al. [45] have reported that the co-treatment of HG and LPS might cause a more intense inflammatory response in cells and organisms compared to HG or LPS treatment alone, suggesting that HG and LPS had a synergistic pro-inflammatory effect on the inflammatory response. 

Crosstalk is an important direction to study the correlation between different signaling pathways. Relevant studies reported that when RAGE and TLR4 were activated, a crosstalk phenomenon occurred between RAGE and TLR4, and the RAGE–TLR4 crosstalk synergistically maintained and amplified the inflammatory response [31,46]. In previous studies, Ayala et al. [47] found that HG can upregulate the protein level of TLR4 on the cell membrane surface, which increases the sensitivity of cells to LPS and enhances the intensity of the initial inflammatory signal after LPS binds to TLR4, thus intensifying the LPS-induced inflammatory response. Nareika et al. [48] found that HG promoted CD14 expression, thereby amplifying the intensity of the initial inflammatory signal stimulated by LPS. The mechanism was that HG significantly upregulated NF-κB and AP-1 activity, thus promoting LPS-induced CD14 expression and inflammatory response. In the present study, LPS + HG treatment further increased the levels of RAGE, TLR4, MyD88, and NF-κB p65 expression and the release of IL-1β, IL-6, and TNF-α. Meanwhile, TAK-242 and FPS-ZM1 blocked the HG + LPS-induced increase in RAGE, TLR4, MyD88, and NF-κB p65 expression levels and the levels of IL-1β, IL-6, and TNF-α. These results suggest that HG and LPS exert synergistic pro-inflammatory effects in BAMs through the RAGE/TLR4/MyD88/NF-κB p65 pathway. Interestingly, the inhibition of RAGE could down-regulate TLR4 expression levels, and the inhibition of TLR4 could down-regulate RAGE expression levels in the presence of HG and LPS co-treatment in BAMs. These results indicate that RAGE–TLR4 crosstalk plays an important role in the activation of the RAGE/TLR4/MyD88/NF-κB inflammatory signaling pathway and the release of pro-inflammatory cytokines IL-1β, IL-6, and TNF-α in the inflammatory response of BAMs caused by HG and LPS co-treatment. Consistent with our results, previous studies have demonstrated that different ligands, such as S100A8/A9, LPS, and HMGB1 can cause RAGE–TLR4 crosstalk, which can further upregulate NF-κB expression as well as pro-inflammatory cytokine release to cause more severe inflammatory responses [46,49,50]. Collectively, when HG and LPS co-treatment-induced inflammation occurred, the TLR4/RAGE levels on the surface of BAMs increased and elevated levels of TLR4/RAGE via RAGE–TLR4 crosstalk. This further increased the possibility of binding with the gram-negative bacteria derived LPS, HG-derived HGBM1, and S100A8/A9 in the synergistic activation of downstream MyD88/NF-κB inflammatory signaling pathways, thus promoting the inflammation response in BAMs. However, there are still many gaps in the research on the mechanisms by which HG exacerbates the inflammatory response of cells induced by LPS. Although HG does not bind to RAGE or TLR4 directly, it can induce the release of pro-inflammatory mediators such as HMGB1 and AGEs and upregulate RAGE and TLR4 expression through the oxidative stress pathway. This, in turn, leads to the binding of mediators to RAGE or TLR4 and the subsequent activation of downstream inflammatory signaling pathways [23,31]. However, existing studies tend to ignore the process of RAGE or TLR4 activation by HG and focus on the changes in downstream signaling pathways after RAGE or TLR4 activation by HG. The crosstalk mechanism involved in this study is also included in this list. Subdividing the pro-inflammatory process of HG to investigate the involvement of additional HG-related pro-inflammatory mediators or pathways in RAGE and TLR4 activation, as well as examining the impact of HG-related pro-inflammatory mediators such as HMGB1 and AGEs on the promotion of inflammatory response by LPS through RAGE/TLR4, instead of treating the HG pro-inflammatory process as a whole, could provide further insight into the molecular mechanisms by which HG exacerbates the inflammatory response of cells induced by LPS. In addition, whether pro-inflammatory mediators produced by HG through oxidative stress such as HMGB1, AGEs, and S100A8/A9 can compete with LPS to bind RAGE and TLR4, and whether they have antagonistic effects with LPS needs to be further investigated. The possible mechanisms of RAGE–TLR4 crosstalk in the inflammatory response of BAMs caused by LPS and HG co-treatment are illustrated in Figure 6.

## 4. Materials and Methods

### 4.1. Isolation and Treatment of BAMs

BAMs were obtained from the intact bovine lungs of five healthy Chinese Simmental cattle (400–500 kg b.w., male, the lung showed no signs of bacterial infection) from a local slaughterhouse, and the isolation protocol was performed following the previous description [51]. Briefly, the lungs were lavaged with approximately 2 L of high-glucose Dulbecco’s modified Eagle medium (DMEM) (Solarbio, Beijing, China) 3 times, and the irrigation fluid was filtered by 200 mesh sterile gauze to remove tissues. Cells were washed with PBS three times and then were cultured in DMEM supplemented with 10% fetal bovine serum (FBS) (ThermoFisher, Shanghai, China), 1% penicillin (100 U/mL), and streptomycin (100 mg/mL) (Gibco, Shanghai, China). Cells were cultured with 5% carbon dioxide (CO_2_) at 37 °C. The non-adherent cells were removed 3 h later by PBS washing. Then, the remaining cells were digested, resuspended, and adjusted to 1 × 10^6^ per mL. The cells were dispensed into T25 cell culture flasks and incubated for 3 h to allow re-adherence, pending subsequent processing.

In the LPS treatment experiments, BAMs were treated with (0, 1, 5, 10 μg/mL) LPS (from *Escherichia coli* 055: B5, SIGMA-L6529, St. Louis, MO, USA) for 12 h or with 5 μg/mL LPS for 0, 3, 6, 12, and 24 h. In the combined use of LPS and HG experiments, BAMs were pretreated with inhibitors for 1 h before adding LPS (5 μg/mL), and inhibitors were dissolved in 100% DMSO (Sigma-Aldrich, St. Louis, MO, USA) and then blended in a culture medium. After adding HG or LPS, the same concentration of inhibitor was re-added to the culture medium. Table 1 displays the grouping as well as the processing. FPS-ZM1 and TAK-242 were purchased from MCE, Shanghai, China.

### 4.2. Real-Time Quantitative Polymerase Chain Reaction (RT-qPCR) 

A Trizol reagent was used to extract the total ribonucleic acid (RNA) from each treatment in cultivated BAMs (ThermoFisher, Shanghai, China). Reverse transcription was performed on the isolated total RNA using a kit to create complementary deoxyribonucleic acid (cDNA) (Takara, Tokyo, Japan). The 96-well plate used for the RT-qPCR contained 4.0 μL diluted cDNA, 5.0 μL of TaKaRa SYBR Green PCR MIX, and 0.5 μL of upstream and downstream primers. For RT-qPCR, the following amplification procedures were used: 95 °C pre-denaturation for 3 min, 95 °C for 10 s; 58 °C for 30 s, and 40 cycles. Using a Real-Time PCR Kit (Takara, Tokyo, Japan), the CFX96 Touch Real-Time PCR Detection System was used to perform the RT-qPCR experiments (Bio-Rad, CA, USA). Table 2 displays the primer sequences that were utilized. The relative gene expression level was estimated using the 2^−ΔΔCt^ method with β-actin as a standard to normalize the RT-qPCR data. 

### 4.3. Immunoblot Assay

The cultured BAMs from each treatment were collected, and the total proteins were extracted using the RIPA lysis buffer. After centrifuging the lysates, the supernatants were analyzed using the BCA Protein Assay Kit (Beyotime, Shanghai, China). Each sodium dodecyl sulfate-polyacrylamide gel (10%) lane was filled with identical quantities of protein (20 μL), and the gels were electrophoresed to separate them. Using the Bio-Rad Trans-Blot device, the separated proteins were subsequently transferred through an electron transfer procedure to polyvinylidene fluoride membranes (Bio-Rad, Hercules, CA, USA). Next, 5% skim milk was used to soak the polyvinylidene fluoride (PVDF) membranes before blocking. They were then incubated with RAGE antibody (ab37647, Abcam, 1:1000), TLR4 antibody (19811-1-AP, Proteintech 1:1000), MyD88 antibody (70R-50098, Fitzgerald 1:1000), NF-κB p65 (C22B4, CST, 1:1000), and β-actin antibody (4970, CST, 1:5000) overnight at 4 °C. The membranes were then incubated with HRP anti-rabbit secondary antibodies (SE134, Solarbio, 1:5000) for 1 h with moderate shaking. With the use of an enhanced chemiluminescence (ECL) kit, blots were seen (Thermo Scientific, Rockford, UK). With the help of the Tanon-5200 automated chemiluminescence imaging analysis system, the band intensity of the pictures was assessed (Tanon, Shanghai, China). Image-Pro Plus 6.0 software was used to examine the integrated optical density (IOD) of each protein band (Media Cybernetics, Inc., Rockville, MD, USA).

### 4.4. Enzyme-Linked Immunosorbent Assay (ELISA)

Isolated BAMs were seeded in a 96-well plate at 1 × 10^5^ cells per well and allowed to settle for 3 h at 37 °C in a 5% CO_2_ incubator. Then, the levels of pro-inflammatory cytokines IL-1β, IL-6, and TNF-α in the supernatants of cultured BAMs in different groups (grouping processing steps as stated in Section 4.1) were measured using ELISA kits (double-antibody sandwich method). All measurements were performed following the manufacturer’s instructions (Jingmei Biotechnology, Jiangsu, China). All experiments were performed three times independently, once in triplicate.

### 4.5. Statistical Analysis

All experiments were performed in triplicate, and the test results are expressed as mean ± standard deviation (SD). The experimental data were sorted and unified in Excel, and experimental data visualization was performed using GraphPad Prism 9.0 (GraphPad Software, CA, USA). The SPSS 26.0 software (IBM, New York, NY, USA) was used to carry out a One-way analysis of variance (ANOVA) test, followed by a least significant difference (LSD) post hoc test. *p* < 0.05 and *p* < 0.01 indicate statistical significance.

## 5. Conclusions

In conclusion, this study demonstrated that RAGE–TLR4 crosstalk was critical in the HG-enhanced LPS-induced inflammation response in BAMs. LPS and HG enhance the inflammatory response of BAMs not only through activation of RAGE and TLR4 but also through the crosstalk between RAGE and TLR4. The crosstalk between RAGE and TLR4 can regulate the genes’ transcription and translation at the receptor level and synergistically activate the downstream MyD88/NF-κB inflammatory signaling pathway that enhances the synthesis and release of pro-inflammatory cytokines TNF-α, IL-1β, and IL-6 in BAMs. These results highlighted the potential of disrupting the crosstalk between RAGE and TLR4 to mitigate the cellular inflammatory response triggered by HG and LPS, emphasizing that simultaneous induction of inflammation responses by multiple pro-inflammatory factors should be avoided. 

## Figures and Tables

**Figure 1 ijms-24-07007-f001:**
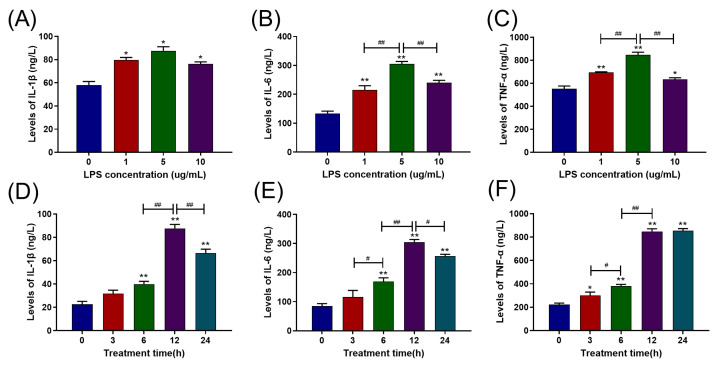
Effects of different LPS treatment times/doses on pro-inflammatory cytokine release in BAMs. (**A**–**C**) Effects of various LPS concentrations (0, 1, 5, and 10 μg/mL) on the release of pro-inflammatory cytokines IL-1β, IL-6, and TNF-α in BAMs. (**D**–**F**) Effects of different LPS treatment times (0, 3, 6, 12, and 24 h) on the release of pro-inflammatory cytokines IL-1β, IL-6, and TNF-α in BAMs. Each experiment was carried out at least three times, and the data were displayed using mean ± SD. *, *p* < 0.05; **, *p* < 0.01 vs. control group; ^#^, *p* < 0.05; ^##^, *p* < 0.01 vs. other treatment groups; One-way ANOVA.

**Figure 2 ijms-24-07007-f002:**
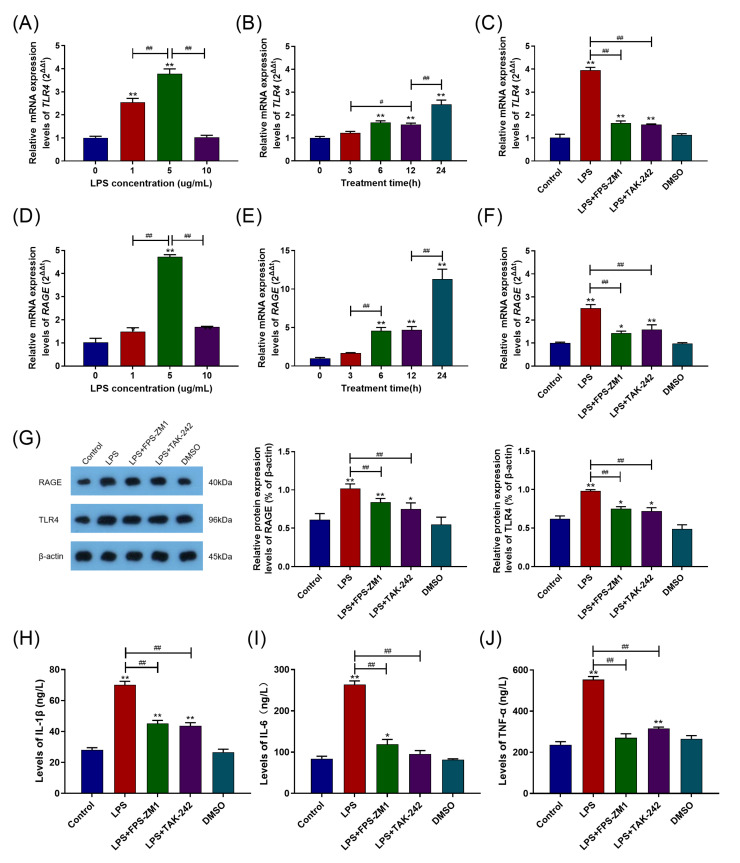
TLR4, RAGE and their interaction were involved in LPS-induced inflammatory response in BAMs. (**A**,**D**) Effects of different LPS concentrations (0, 1, 5, and 10 μg/mL) on the levels of TLR4 and RAGE mRNA expression. (**B**,**E**) Effects of different LPS treatment time (0, 3, 6, 12, and 24 h) on the levels of TLR4 and RAGE mRNA expression. (**C**,**F**) Effects of pretreatment with FPS-ZM1 and TAK-242 on the levels of TLR4 and RAGE mRNA expression. (**G**) Comparison of gel images and grayscale for RAGE and TLR4 immunoblot detection. (**H**–**J**) The levels of IL-1β, IL-6, and TNF- α in the supernatants of each group. Each experiment was carried out at least three times, and the data were displayed using mean ± SD. *, *p* < 0.05; **, *p* < 0.01 vs. control group; ^#^, *p* < 0.05; ^##^, *p* < 0.01 vs. other treatment groups; One-way ANOVA.

**Figure 3 ijms-24-07007-f003:**
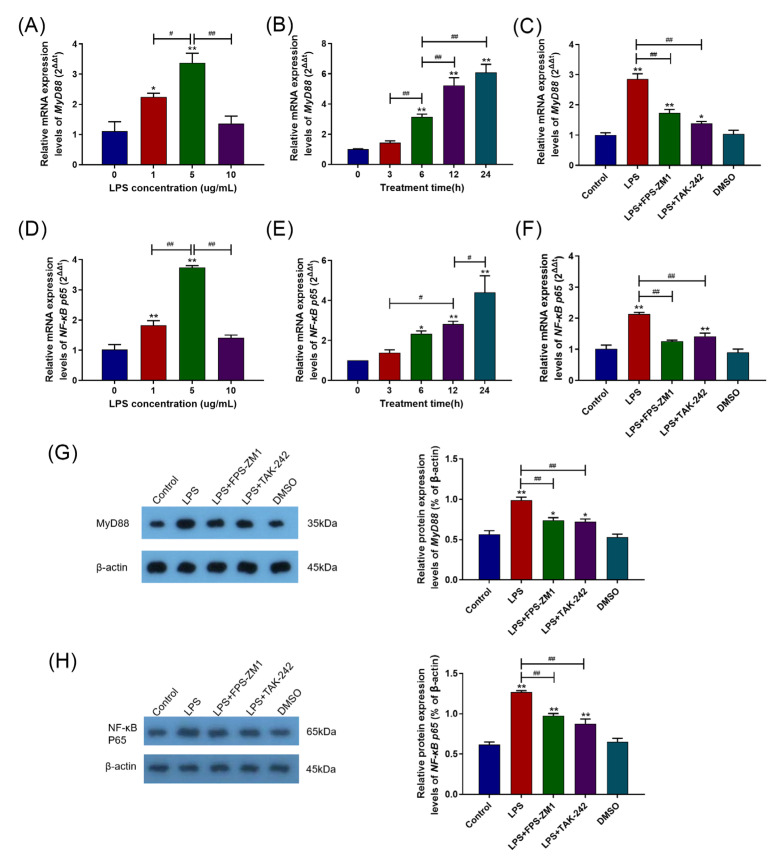
RAGE and TLR4 synergistically activated the *MyD88*/*NF-κB* signaling pathway in the inflammation caused by LPS. (**A**,**D**) Effects of different LPS concentrations (0, 1, 5, and 10 μg/mL) on the levels of MyD88 and NF-κB p65 mRNA expression. (**B**,**E**) Effects of different LPS treatment times (0, 3, 6, 12, and 24 h) on the levels of MyD88 and NF-κB p65 mRNA expression. (**C**,**F**) Effects of FPS-ZM1 and TAK-242 pretreatment on the levels of MyD88 and NF-κB p65 mRNA expression. (**G**,**H**) Comparison of gel images and grayscale for MyD88 and NF-κB p65 immunoblot detection. Each experiment was carried out at least three times, and the data were displayed using mean ± SD. *, *p* < 0.05; **, *p* < 0.01 vs. control group; *^#^*, *p* < 0.05; *^##^*, *p* < 0.01 vs. other treatment groups; One-way ANOVA.

**Figure 4 ijms-24-07007-f004:**
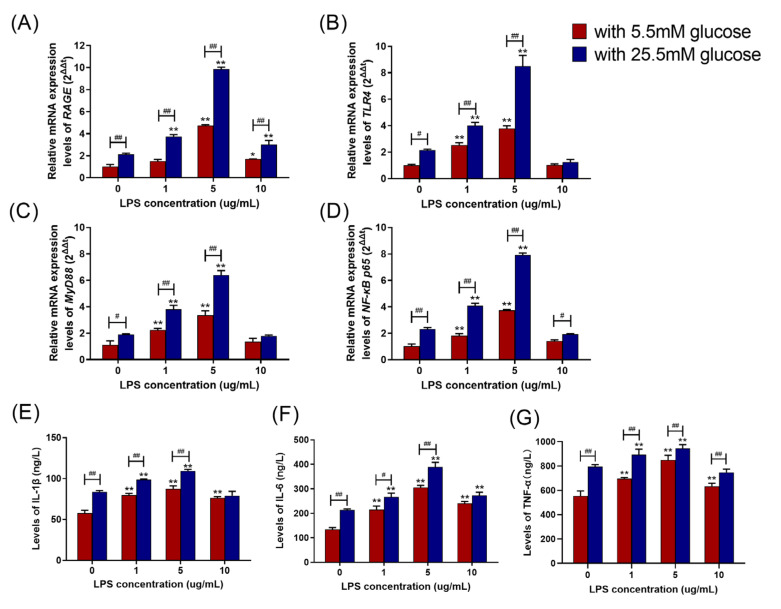
Effects of different glucose concentrations on LPS-induced pro-inflammatory cytokine secretion and the *RAGE*/*TLR4*/*MyD88*/*NF-κB p65* pathway in BAMs. (**A**–**D**) The impact of NG and HG on the levels of RAGE, TLR4, MyD88, and NF-κB p65 mRNA expression in the presence of different concentrations of LPS (0, 1, 5, and 10 μg/mL). (**E**–**G**) The impact of NG and HG on the levels of IL-1β, IL-6, and TNF-α in the presence of various concentrations of LPS (0, 1, 5, and 10 μg/mL). Each experiment was carried out at least three times, and the data were displayed using mean ± SD.*, *p* < 0.05; **, *p* < 0.01 vs. control group; *^#^*, *p* < 0.05; *^##^*, *p* < 0.01 vs. other treatment groups; One-way ANOVA.

**Figure 5 ijms-24-07007-f005:**
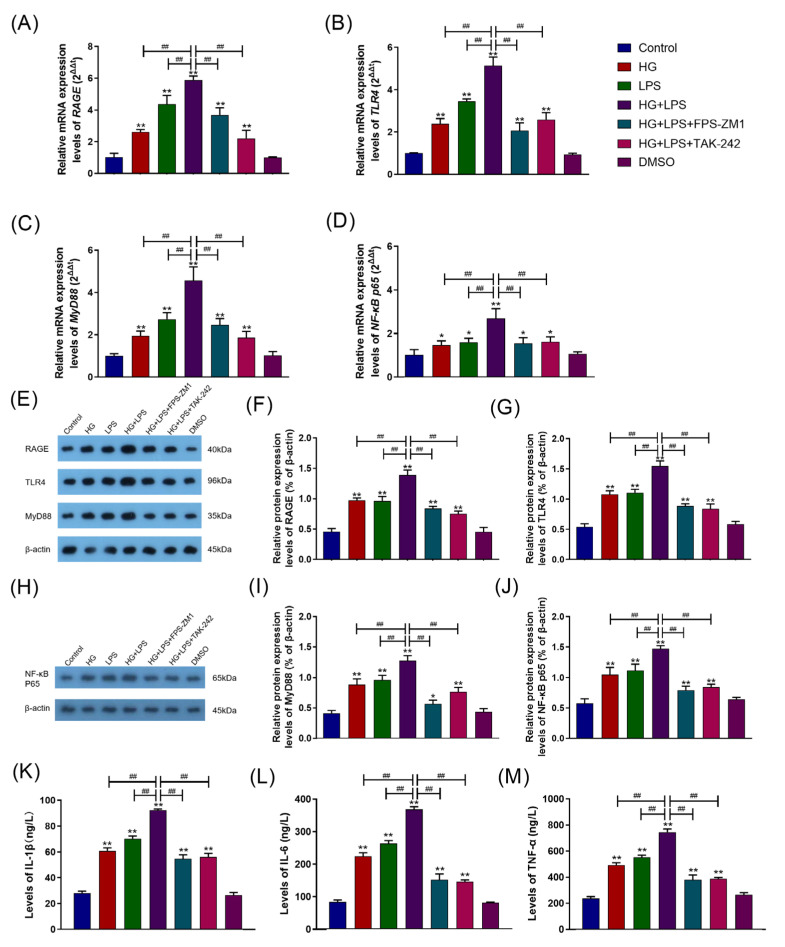
RAGE–TLR4 crosstalk regulated the synergism between high glucose and LPS on the inflammatory response in BAMs. (**A**–**D**) The levels of RAGE, TLR4, MyD88 and NF-κB p65 mRNA expression in each group. (**E**–**J**) The levels of RAGE, TLR4, MyD88 and NF-κB p65 protein expression in each group. (**K**–**M**) The levels of IL-1β, IL-6, and TNF-α in each group. Each experiment was carried out at least three times, and the data were displayed using mean ± SD. *, *p* < 0.05; **, *p* < 0.01 vs. control group; *^##^*, *p* < 0.01 vs. other treatment groups; One-way ANOVA.

**Figure 6 ijms-24-07007-f006:**
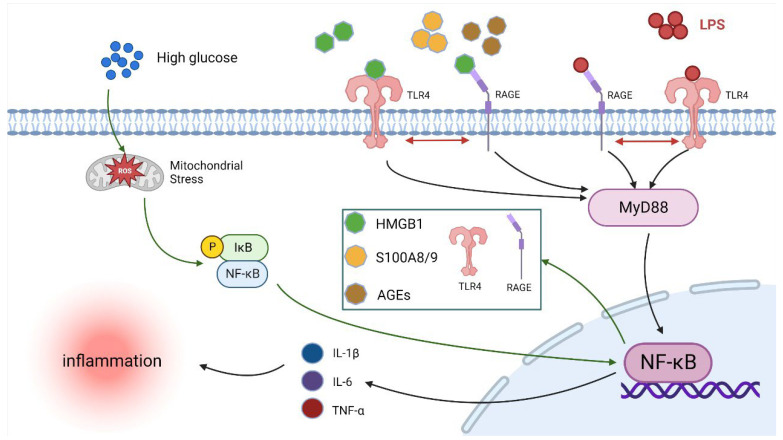
Schematic representation of the TLR4–RAGE crosstalk participates in the HG-enhanced LPS-induced inflammation in BAMs (created with BioRender.com). HG firstly upregulated the levels of RAGE and TLR4 genes and protein expression in the BAM membrane and the secretion of pro-inflammatory-related mediators (such as HMGB1, S100A8/9, and AGEs) through the ROS/NF-κB pathway. Then, pro-inflammatory-related mediators bind to RAGE and TLR4, causing RAGE–TLR4 crosstalk (increasing the levels of RAGE and TLR4) to synergistically activate the downstream MyD88/NF-κB signaling pathway and promote the release of pro-inflammatory cytokines IL-1β, IL-6, and TNF-α. When HG acts together with LPS, HG upregulates the levels of RAGE and TLR4 genes and protein expression, which can provide more receptor sites for LPS binding, and the LPS also upregulates the levels of RAGE and TLR4 genes and protein expression that can offer more receptor sites for pro-inflammatory-related mediators. The combination of HG and LPS through RAGE–TLR4 crosstalk further activates the downstream MyD88/NF-κB signaling pathway and exacerbates pro-inflammatory cytokine release in BAMs.

**Table 1 ijms-24-07007-t001:** Experimental grouping of the combined effects of LPS and HG.

Group	Treatment
Control	0 μg/mL LPS + 5.5 mM glucose
HG	0 μg/mL LPS + 25.5 mM glucose
LPS	5 μg/mL LPS + 5.5 mM glucose
HG +LPS	5 μg/mL LPS + 25.5 mM glucose
HG + LPS + FPS-ZM1	5 μg/mL LPS + 25.5 mM glucose +1 μM FPS-ZM1
HG + LPS + TAK-242	5 μg/mL LPS + 25.5 mM glucose +10 μM TAK-242
DMSO	0 μg/mL LPS + 5.5 mM glucose + DMSO

**Table 2 ijms-24-07007-t002:** The primer sequence of genes for Real-Time PCR.

Gene Name	Forward Primer Sequence (5′–3′)	Reverse Primer Sequence (3′–5′)
*β-actin*	GCCCATCTATGAGGGGTACG	TCACGGACGATTTCCGCT
*RAGE*	GACAGTCGCCCTGCTCATT	CCTCTGGCTGGTTCAGTTCC
*TLR4*	TGCCTTCACTACAGGGACTTT	TGGGACACCACGACAATAAC
*NF-κB p65*	GAGATCATCGAGCAGCCCAA	ATAGTGGGGTGGGTCTTGGT
*MyD88*	AGAAGAGGTGCCGTCGGATGG	TTGGTGTAGTCACAGACAGTGATGAAG

## Data Availability

The data used to support the findings of this study are available from the corresponding author upon request.

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
