# Peer review of "RAGE–TLR4 Crosstalk Is the Key Mechanism by Which High Glucose Enhances the Lipopolysaccharide-Induced Inflammatory Response in Primary Bovine Alveolar Macrophages"

_ijms, 2023, doi:10.3390/ijms24087007_

Round 1

Reviewer 1 Report

The authors verified that RAGE and TLR4 affect each other's expression through crosstalk during the combined usage of HG and LPS, and synergistically activate the MyD88/NF-κB signaling pathway to promote the release of pro-inflammatory cytokines in BAMs. Some important points must be clarified or fixed before we can proceed, and an action can be taken:

1. The levels of TLR4 and RAGE protein expression in figure 2G showed no significant difference after the treatment with FPS-ZM1 and TAK-242. Please clarify.

2. The authors are suggested to examine the expression level of phospho-p65 using western blot after the treatment with FPS-ZM1 and TAK-242.

3. Why did the authors select MyD88/NF-κB signaling pathway to verify? How about the alterations of other signaling pathways?

4. Each discussion presented is intriguing, but overall, the work falls short of demonstrating molecular biology at the inflammatory level on the phenotypes observed. Indeed, it is equally possible that the identified genes act on the inflammatory environment indirectly since there is no competitive binding of HG and LPS to TLR4 or RAGE, and thus the biological function in these changes needs to be explored in much more detail.

5. All the immunoblotting bands were very similar, and highly suspect that the data were fabricated in this study. There are large molecular weight proteins, as well as small proteins. The bands of these proteins could not be that similar and have no heterobands. Please explain.

6. The authors make certain that the statistical analysis is appropriate, and that it is accurately described in the manuscript.

Author Response

Thank you very much for your comments and advice, and your advice increased the quality of our manuscript very much. According to your suggestion, we’ve revised our manuscript carefully, and the responses are as follows:

Comment 1: The levels of TLR4 and RAGE protein expression in figure 2G showed no significant difference after the treatment with FPS-ZM1 and TAK-242. Please clarify.

Response 1: Thanks for your comment. We have revisited our raw data and are confident that the levels of TLR4 and RAGE protein expression in figure 2G showed no significant difference after the treatment with FPS-ZM1 and TAK-242. We apologize for confusing you because we did not adequately describe the results and the description of the results has been added (lines 152-155). In this part of our study, we aimed to investigate whether TLR4, RAGE, and their interaction were involved in the inflammatory response caused by LPS in BAMs. It is known that TAK-242 inhibits TLR4 expression and FPS-ZM1 inhibits RAGE expression. However, in figure 2G, our results demonstrate that the protein levels of RAGE decreased when TLR4 expression was inhibited with TAK-242, and the protein levels of TLR4 decreased when RAGE expression was inhibited with FPS-ZM1, suggesting the crosstalk between RAGE and TLR4, and this result is consistent with the purpose of our study. We thank the reviewers for bringing this to our attention and appreciate the opportunity to improve our manuscript.

Comment 2: The authors are suggested to examine the expression level of phospho-p65 using western blot after the treatment with FPS-ZM1 and TAK-242.

Response 2: We appreciate your suggestion to examine the expression level of phospho-p65 using a western blot after treatment with FPS-ZM1 and TAK-242. However, our study focused on investigating the role of RAGE-TLR4 crosstalk in HG enhancement of LPS-induced inflammatory responses in BAMs, including enhanced activation of downstream MyD88/NF-κB signaling pathway and release of pro-inflammatory cytokines TNF-α, IL-1β, and IL-6 instead of just focusing on NF-κB pathway. NF-κB p65, as a common diagnostic marker for inflammatory diseases, is activated downstream of TLR4 and RAGE signaling and regulates the expression of genes involved in inflammation and immunity. Testing for NF-κB p65 can help determine if NF-κB is being activated in inflammatory diseases. Therefore, the detection of NF-κB p65 is sufficient to justify our conclusion that crosstalk between RAGE and TLR4 plays a critical role in the inflammatory response induced by the combination of HG and LPS in BAMs. We sincerely appreciate your feedback and hope that this explanation addresses your concern.

Comment 3: Why did the authors select MyD88/NF-κB signaling pathway to verify? How about the alterations of other signaling pathways?

Response 3: Thank you for your comment on our manuscript. In our study, we selected the MyD88/NF-κB signaling pathway to verify because it is a well-known and critical pathway involved in the regulation of pro-inflammatory cytokines, such as TNF-α, IL-1β, and IL-6. Moreover, MyD88/NF-κB signaling pathway is a common downstream pathway of TLR4 and RAGE, which is in line with our study objective that TLR4 and RAGE synergistically activate the downstream MyD88/NF-κB signaling pathway through a crosstalk mechanism to enhance the inflammatory response. Additionally, our study aimed to investigate the crosstalk between RAGE and TLR4 in the activation of the MyD88/NF-κB signaling pathway and the subsequent release of pro-inflammatory cytokines. This crosstalk has been reported in several previous studies, and we aimed to confirm and extend these findings in the combined effect of LPS and high glucose. While other signaling pathways may also be involved in the regulation of pro-inflammatory cytokines, such as MAPK and JAK-STAT signaling pathways, our focus was on the RAGE/TLR4/MyD88/NF-κB signaling pathway as it is a central pathway in the immune response and inflammation. In future studies, it would be interesting to investigate the effect of TAK-242 and FPS-ZM1 on these other signaling pathways as well, to obtain a more complete understanding of their mechanism of action. We appreciate your feedback and hope that this explanation clarifies our rationale for selecting the MyD88/NF-κB signaling pathway for our study.

Comment 4: Each discussion presented is intriguing, but overall, the work falls short of demonstrating molecular biology at the inflammatory level on the phenotypes observed. Indeed, it is equally possible that the identified genes act on the inflammatory environment indirectly since there is no competitive binding of HG and LPS to TLR4 or RAGE, and thus the biological function in these changes needs to be explored in much more detail.

Response 4: Thank you for your comment on our manuscript. We have revised the discussion section of our manuscript. In the Discussion section we added the mechanisms of HG enhanced LPS-induced inflammatory responses explored in the studies of others to highlight the novelty of the RAGE-TLR4 crosstalk mechanism in HG enhanced LPS-induced inflammatory responses in our study (lines 317-324). Then, we added a discussion on the pathway of HG activation of RAGE and TLR4 through indirect action and added a discussion of the gaps in the current study of HG enhancement of LPS leading to the inflammatory response (lines 348-357). Finally, we added a discussion of the next direction of research, which is not only to study the HG pro-inflammatory process as a whole but to subdivide the HG pro-inflammatory process, which could provide further insight into the molecular mechanisms by which HG exacerbates the LPS-induced cellular inflammatory response (lines 357-366). We will consider conducting further experiments to address these points and provide a more comprehensive understanding of the mechanisms underlying the observed changes. We thank you for your valuable feedback, which will help us improve the quality of our manuscript.

Comment 5: All the immunoblotting bands were very similar, and highly suspect that the data were fabricated in this study. There are large molecular weight proteins, as well as small proteins. The bands of these proteins could not be that similar and have no heterobands. Please explain.

Response 5: Thanks for your comment. We guarantee that the immunoblotting bands in the manuscript are our authentic and unmodified results. All protein immunoblotting tests were performed using high standards of experimental practice (our experimental procedure and the experimental reagents and instruments used are shown in lines 426-442 in the manuscript). Our study aimed to demonstrate that HG promotes the inflammatory response induced by LPS on BAMs through a RAGE-TLR4 crosstalk mechanism. HG acting together with LPS on BAMs resulted in increased protein expression of RAGE/TLR4, which, through a crosstalk mechanism between the two, led to increased protein expression of TLR4/RAGE as well. We used the same batch of cell samples with the same treatment under the same experimental conditions, as well as the fact that both RAGE and TLR4 are surface receptors of cell membranes, while there was crosstalk between RAGE and TLR4. Therefore, the expression trends of RAGE and TLR4 showed similarity. On the other hand, after the protein samples were electrophoresed by SDS-PAGE, the large molecular weight proteins were separated from the small molecular weight proteins, and the target proteins were transferred to the PVDF membrane after current stimulation and formed the immunoblotting bands in the manuscript after color development as well as photography. Therefore, the protein blot on the immunoblotting bands can only represent the expression amount of the target protein, and there is no direct relationship with the molecular weight of the target protein. We have conducted multiple tests using proven testing techniques and rigorous testing procedures. The absence of heterobands in the immunoblotting bands may be due to the high concentration of primary antibodies (the antibodies we use have been validated in multiple trials) we used and the high specificity of binding to the target protein, followed using skim milk enclosed at low temperature for a long time, hence the high auto-exposure intensity of the camera, which reduced the generation of stray bands in the immunoblotting bands. If you have any specific concerns or suggestions regarding our immunoblotting bands, we would be happy to address these issues in more detail.

Comment 6: The authors make certain that the statistical analysis is appropriate, and that it is accurately described in the manuscript.

Response 6: Thanks for your comment. We have rewritten the Statistical Analysis section (lines 455-460) and ensured that the statistical analysis is appropriate and accurately described in the manuscript. We also have made some additions to the graphical legend of the results (lines 167, 192, 218, 248). If you have any further questions or concerns about our statistical analysis, we would be happy to address them in more detail.

Reviewer 2 Report

This is relevant study indicating that interference RAGE-TLR4 crosstalk may attenuate the HG and LPS induced inflammatory response. However, I have some comments that need to be adressed:

1.     Line 89 90 91 92 This sentence should be rewritten as it is something that will be investigated in the study.

2.     The last sentence in the Introduction part should be stated in the conclusion at the end of Discussion part.

3.     Enzyme-Linked Immunosorbent Assay is not sufficiently described.

4.     How do you explain that TAK-242 and FPS-ZM1 significantly decreased the levels of MyD88 and NF-κB p65 expression?

5.     please correct typos (there should be space before square brackets in references)

6.     In Result section y-axes in the graphs is not aligned with the same measure grid, so it is difficult to follow results. This should be aligned as much as possible.

Author Response

Thank you very much for your comments and advice, and your advice increased the quality of our manuscript very much. According to your suggestion, we’ve revised our manuscript carefully, and the responses are as follows:

Comment 1: Line 89 90 91 92 This sentence should be rewritten as it is something that will be investigated in the study.

Response 1: We appreciate your feedback and thank you for helping us to improve the clarity and accuracy of our manuscript. We apologize for any confusion that may have been caused by the wording of this sentence. Based on your suggestion, we have rewritten the sentence in lines 89 90 91 92. The rewritten sentence is in lines 89-97.

Comment 2: The last sentence in the Introduction part should be stated in the conclusion at the end of Discussion part.

Response 2: We appreciate your feedback and according to your advice, we have stated the last sentence of the introductory section in the conclusion section (lines 490-492). Thank you again for your input, and please let us know if you have any further questions or concerns.

Comment 3: Enzyme-Linked Immunosorbent Assay is not sufficiently described.

Response 3: Thanks for your feedback. According to your advice, we have reorganized the sentences and described the ELISA assay in as much detail as possible (lines 457-463). We appreciate your comments and are committed to ensuring that our methods and results are accurately described in the manuscript.

Comment 4: How do you explain that TAK-242 and FPS-ZM1 significantly decreased the levels of MyD88 and NF-κB p65 expression?

Response 4: Thanks for your question.TAK-242 and FPS-ZM1 are known to be inhibitors of Toll-like receptor 4 (TLR4) and receptors for advanced glycation end products (RAGE), respectively. MyD88/NF-kB pathway is a common downstream pathway of RAGE and TLR4. MyD88 is an adaptor protein that is recruited to the TLR4 receptor upon activation, and it plays a critical role in TLR4 signaling. In addition, RAGE activation in macrophages led to the activation of MyD88 and downstream signaling molecules, resulting in the production of pro-inflammatory cytokines. NF-κB p65 is a subunit of the transcription factor NF-κB, which is activated downstream of TLR4 and RAGE signaling and regulates the expression of genes involved in inflammation and immunity. TLR4 and RAGE activate the downstream MyD88 signaling pathways, which ultimately lead to the activation of the transcription factor NF-κB. By inhibiting TLR4 and RAGE, TAK-242 and FPS-ZM1 prevent the activation of MyD88, which in turn prevents the activation of NF-κB p65 and the downstream inflammatory response. On the other hand, our experiments demonstrate that RAGE and TLR4 can influence each other's expression at the receptor level. Thus, when TLR4/RAGE activation is inhibited, it also inhibits RAGE/TLR4 activation, thus affecting the expression of downstream pathway genes. Therefore, TAK-242 and FPS-ZM1 significantly decrease the levels of MyD88 and NF-κB p65 expression by inhibiting the activation of TLR4 and RAGE, attenuating the signal transduced to the downstream pathway and thus reducing the levels of MyD88 and NF-κB p65 expression.

Comment 5: please correct typos (there should be space before square brackets in references)

Response 5: Thank you very much for pointing out the problem. We have corrected the reference style in the manuscript so that it has a space before the square brackets. Thank you again for your feedback.

Comment 6: In Result section y-axes in the graphs is not aligned with the same measure grid, so it is difficult to follow results. This should be aligned as much as possible.

Response 6: Thanks for your feedback. We apologize for any confusion caused by the y-axes in the graphs is not aligned with the same measure grid. We agree that it is important to align the y-axes in the graphs with the same measure grid to facilitate the comparison of the results. We have reviewed all the graphs in the manuscript and made the necessary adjustments to align the y-axes in the graphs with the same measure grid as much as possible (lines 160, 185, 213, and 250). Thank you again for bringing this issue to our attention. 

Round 2

Reviewer 1 Report

In the revised article, the authors modified the manuscript and figures referred to the comments, and answered the questions comprehensively.